# Incidence of malignant transformation in the oviductal fimbria in laying hens, a preclinical model of spontaneous ovarian cancer

**Elizabeth A. Paris**[1], **Janice M. Bahr**[2], **Pincas Bitterman**[3,4], **Sanjib Basu**[5], **Jacques S. Abramowicz**[6], **Animesh Barua**[1,3,4]*

**1** Department of Cell and Molecular Medicine, Rush University Medical Center, Chicago, Illinois, United States of America, **2** Department of Animal Sciences, University of Illinois at Urbana-Champaign, Urbana, Illinois, United States of America, **3** Department of Pathology, Rush University Medical Center, Chicago, Illinois, United States of America, **4** Department of Obstetrics and Gynecology, Rush University Medical Center, Chicago, Illinois, United States of America, **5** Department of Internal Medicine, Rush University Medical Center, Chicago, Illinois, United States of America, **6** Department of Obstetrics and Gynecology, University of Chicago, Chicago, Illinois, United States of America

* Animesh_Barua@rush.edu

**Data Availability Statement:** All relevant data are within the manuscript and its Supporting Information files.

## Abstract

Ovarian high grade serous carcinoma (HGSC) is a lethal form of ovarian cancer (OVCA). In most cases it is detected at late stages as the symptoms are non-specific during early stages. Emerging information suggests that the oviductal fimbria is a site of origin of ovarian HGSC. Currently available tests cannot detect ovarian HGSC at early stage. The lack of a preclinical model with oviductal fimbria that develops spontaneous ovarian HGSC is a significant barrier to developing an early detection test for this disease. The goal of this study was to examine if the oviductal fimbria in hens is a site of origin of HGSC and whether it expresses several putative markers expressed in ovarian HGSC in patients. A total of 135 laying hens (4 years old) were selected from a flock using transvaginal ultrasound (TVUS) imaging, followed by euthanasia and gross examination for the presence of solid masses and ascites. Histological types of carcinomas were determined by hematoxylin and eosin staining. Expression of WT-1, mutant p53, CA-125, PAX2 and Ki67 in normal or malignant fimbriae or ovaries were examined using immunohistochemistry, immunoblotting and gene expression assays. This study detected tumors in oviductal fimbriae in hens and routine staining revealed ovarian HGSC-like microscopic features in these tumors. These tumors showed similarities to ovarian HGSC in patients in expressing several markers. Compared with normal fimbriae, intensities of expression of WT-1, mutant p53, CA-125, and Ki67 were significantly ($P<0.05$) higher in fimbrial tumors. In contrast, expression of PAX2 decreased gradually as the tumor progressed to late stages. The patterns of expression of these markers were similar to those in ovarian HGSC patients. Thus, tumors of the oviductal fimbria in hens may offer a preclinical model to study different aspects of spontaneous ovarian HGSC in women including its early detection.

**Funding:** This study was supported by NIH R01 CA210370 (AB). The funders had no role in study design, data collection and analysis, preparation of manuscript and decision to publish.

## Introduction

Due to the non-specificity of symptoms at early stages, aggressive rates of growth, and the lack of an effective early detection test, high-grade serous carcinoma (HGSC) of the ovary, a fatal form of epithelial ovarian cancer (OVCA), in most cases is detected at late stages [1]. The 5-year survival rates of OVCA patients decrease to less than 30% when the disease is detected at late stages as opposed to more than 90% when detected at early stages [2]. Moreover, tumors recur frequently as they develop resistance against currently available therapeutics [3]. Thus, early detection of ovarian HGSC remains critical for improving the rates of survival of patients. Until recently, the ovarian surface epithelium was predicted to be the only site of origin of ovarian HGSC. Emerging information suggests that OVCA is a disease with heterogeneous origin and the fimbria of the fallopian tube has also shown as a site of origin of ovarian HGSC [4]. Unfortunately, difficulty in identifying patients at early stages of ovarian HGSC and access to fimbrial tissues are significant barriers to the study and development of an effective early detection test for ovarian HGSC.

Animal models have long been used to understand the pathophysiology of human diseases, as well as for development of tests for their detection and interventional strategies which are difficult to study in humans [5, 6]. For ovarian HGSC, rodent models have been used extensively to generate information despite not being a spontaneous model [7]. Several rodent models of induced OVCA have been developed either chemically, including hormonal or non-hormonal, genetically, including knockout or transgenic, or with xenograft or syngeneic methods [6, 8]. The genetically engineered mouse models (GEMMs) of HGSC have reported to develop HGSC with histopathologic features seen in many human HGSCs [9, 10]. Ovariectomized mice have been reported to develop HGSC while removal of fallopian tube prevented its incidence [11]. In addition, mice genetically engineered to harbor *Dicer 1* and *Pten* inactivation and mutant p53 showed the fallopian tube as a site of ovarian HGSC [12].

Unfortunately, these models have some inherent limitations. Following detachment, tubal epithelial cells are implanted in the ovary and the bursa covering the mouse ovary is a physical barrier to such implantation [7]. Endosalpingiosis has been suggested as a precursor for developing OVCA, however, no such extra-ovarian (tubal) precursor is present in these mouse models [7]. Moreover, endometriosis and polycystic ovarian syndrome (PCOS) have also been reported as risk factors for OVCA development [13, 14]. Unfortunately, rodents do not develop these conditions. Although large animals, including bovines, develop OVCA spontaneously, their long generation intervals, low rates of incidence, and the managerial requirements of these animals make them unsuitable to be a preclinical model for the study of fimbria and its malignant transformation leading to ovarian HGSC [7]. Thus, additional animal model (s) which develop OVCA spontaneously and are feasible for the study of fimbrial malignant transformation need to be explored.

Laying hens have been reported to develop OVCA spontaneously with remarkable similarities to OVCA in women, including its dissemination (with the formation of ascites), histological types of tumors, and expression of several key markers [15–17]. In addition, similarities in gene expression between ovarian tumors and normal oviduct have been reported in laying hens [18], suggesting a possibility of oviductal contribution to a portion of ovarian cancer as proposed for women. Moreover, similarities exist between hens and women with regard to the normal endocrine regulation of ovarian functions, such as follicular growth and ovulation [19]. The theory of incessant ovulation as a predisposing factor for the development of OVCA has been supported by the laying hen, model as they are frequent ovulators [19, 20]. Therefore, hens have the potential to be a biologically and clinically relevant model of spontaneous OVCA to examine the etiology, pathophysiology, and progression of OVCA and to develop

diagnostics and therapeutic interventions. Furthermore, the reproductive tract of hens also consists of fimbriae and, as is observed in the fallopian tube of women, the ovulated egg is received by the fimbria of the oviduct in hens following ovulation. Through this process, fimbrial tissues in women and in hens are both exposed to various biomolecules and inflammatory agents present in the follicular fluid [21]. Thus, laying hens may represent a suitable animal model for the study of fimbrial malignant transformation leading to spontaneous ovarian HGSC development if they express markers reported in ovarian HGSC in patients.

WT-1 (Wilms Tumor 1) is a transcription factor and plays an essential role in the normal development of the urogenital system and 93% of ovarian serous carcinomas showed to express nuclear staining for WT-1 [22]. Tumor protein p53, a master regulator and a tumor suppressor protein, also known as guardian of the genome, has been shown to be mutated in more than 96% of cases in ovarian cancer [23]. CA-125 (cancer antigen 125) is a member of the mucin family of glycoproteins [24] and has been shown to play a role in advancing tumorigenesis and tumor proliferation by several different mechanisms. Serum level of CA-125 is used to monitor the progression and/or efficacy of therapies against ovarian cancer [25]. The PAX (paired box) gene PAX2 is a lineage-specific transcription factor that is involved in epithelial development of the fallopian tube but not the ovary [26, 27]. PAX2 has been reported to be expressed in prostate cancer [28] and in both high- and low-grade ovarian tumors in women [29, 30]. As in women, PAX2 expression has also been reported in oviduct of normal hens as well as in ovarian tumors [18]. The goal of this exploratory study was to examine the incidence of malignant transformation in oviductal fimbriae in hens and to determine whether fimbrial tumors in hens also express markers commonly expressed in ovarian HGSC in women including WT-1, p53, CA125, and PAX2.

## Methods & materials

### Hen specimens

Commercial strains of White Leghorn laying hens (approx. 4 years old, $n = 135$) were randomly selected from a flock and scanned using transvaginal ultrasound (TVUS) imaging as reported earlier [31, 32]. Hens were reared at the University of Illinois at Urbana-Champaign (UIUC) Poultry Research Farm in individual cages under a 17-hour light/ 7-hour dark schedule with feed and water provided *ad libitum*. Appropriate hygienic conditions were maintained throughout the environment and egg-laying rates were monitored. Distress and suffering were minimized in all practices. Blood from all hens was collected from the brachial vein (wing vein) and serum was separated and stored in -80˚C for future use. All hens were euthanized via cervical dislocation or inhalation of carbon dioxide, consistent with humane recommendations of the American Veterinary Medical Association, and examined for the presence of any abnormality, including solid tissue mass in the ovary, oviductal fimbria and other parts of the oviduct, as well as any other organs and tissues as reported earlier [15]. Extent of malignancy when present was recorded during gross examination following euthanasia as reported earlier [15]. Fimbriae and/or ovaries with solid mass as well as fimbrial and ovarian tissues from healthy hens were collected. All selected tissues were processed for either paraffin, frozen sections, proteomic or genomic studies. All procedures for animal use were performed according to the protocol (Protocol Number: 19091) approved by the Institutional Animal Care and Use Committee (IACUC) of the University of Illinois at Urbana-Champaign (UIUC).

### Clinical specimens

Archived clinical specimens, including normal ovarian ($n = 5$, 60-65years old) tissues were collected from the Department of Pathology at Rush University Medical Center (RUMC). These

subjects underwent prophylactic surgery due to risk of OVCA development. Specimens from patients with ovarian HGSC ($n = 5$, $n = 60$–70 years old) were also used. Clinical specimens served as the reference to study the resemblance in the expression of markers among hen fimbrial and ovarian tumors to human ovarian HGSC. All clinical specimens were full anonymized prior to this study and used as per the protocols approved by the Institutional Review Board (IRB) of Rush University Medical Center.

## Histopathological examination of specimens

Presence or absence of malignancy in hen fimbrial and/or ovarian tissues and their histological subtypes were determined by routine hematoxylin and eosin staining using paraffin-embedded sections as reported earlier [15]. Diagnosis for clinical specimens including normal tissues or histological types of tumors and their stages were obtained from the final reports from the Department of Pathology at Rush University Medical Center, Chicago, IL.

## Immunohistochemistry (IHC)

Expression of markers in normal or malignant tissues was determined by immunohistochemistry using paraffin sections as reported earlier [33, 34]. Briefly, sections were deparaffinized with xylene and rehydrated using a graded series of ethanol and rinsing in deionized (DI) water. Antigens on the sections were retrieved by heating using citrate solution (pH 6.0), followed by quenching of endogenous peroxidase by incubating with 0.3% $H_2O_2$ in methanol. Non-specific binding of antibodies was blocked by incubating the sections with normal horse serum (Vector Laboratories, Burlingame, CA). Sections were then incubated overnight with primary antibodies, including anti-WT-1, anti-p53 or anti-Ki67 (Millipore Sigma, St. Louis, MO) or anti-CA-125 (Invitrogen, Thermo Fisher Scientific, Waltham, MA) or anti-PAX2 (R&D Systems, Inc., Minneapolis, MN) at a 1:100 dilution. Additional information for each antibody is available in (S1 Table). After incubation, sections were washed with phosphate buffered saline (PBS, 3X5min) and incubated with respective biotinylated secondary antibodies for 1 hour. After washing with PBS (3X5min), sections were incubated with peroxidases conjugated with avidin for 1 hour. Sections were then washed with PBS (3X5min) and immunoreactions on the sections were visualized with 3, 3'Diaminobenzidine (DAB) containing $H_2O_2$ substrate by observing under a light microscope. Once the reaction was complete, sections were washed in DI water, counter-stained with hematoxylin, dehydrated with a graded series of ethanol and xylene, and mounted with organic mounting media and covered with a cover slip and oven dried. Sections were then examined under a light microscope attached to a computer-assisted software program for imaging (MicroSuite™ version 5, Olympus American, Inc., Center Valley, PA). Images from 3–5 areas at a 40X magnification in a section containing the highest immunoreactive cells or with strong staining were taken and archived. Specificity of immunostaining for each antibody was confirmed by control staining of sections in which primary antibodies were omitted. No staining was observed in these sections.

## Counting of immunostaining

Archived images were examined to determine the intensity of immunostaining for each marker, including WT-1 or PAX2. Staining intensities were determined using a computer-assisted software program (MicroSuite™ version 5, Olympus American, Inc., Center Valley, PA). The mean intensity for each marker in a section was determined as the arbitrary values from the 3–5 areas at 40X magnification. Values were converted to be expressed as intensities in 20,000 μm² area of tissue. Two sections were used for each marker from each tissue from each hen in a group. The average of the intensities of all hens in a group were considered as the

mean intensity of each marker in each group (normal fimbria, tumor fimbria, and ovarian tumors). Similarly, frequencies of p53, CA-125, and Ki67 expressing cells among different groups of hens were determined.

## Western blotting

Immunohistochemical expression of WT-1, p53, CA-125, PAX2 and Ki67 were confirmed by immunoblotting of homogenates collected from representative specimens of normal fimbriae, fimbriae with tumors, and ovaries with tumors in hens using the primary antibodies at a 1:1000 dilution for each marker mentioned above as reported previously at [35]. Appropriate horse-radish peroxidase conjugated secondary antibodies were used and immunoreactions on the membranes were detected as chemiluminescence products and imaged by ChemiDoc XRS (Bio-Rad Laboratories, Hercules, CA). Images were archived for later use.

## Gene expression studies

Expression of WT-1, p53, CA-125, PAX2, and Ki67 genes in normal fimbriae, fimbriae with tumors, and ovaries with tumor were examined by reverse-transcriptase polymerase chain reaction (RT-PCR) and quantitative RT-PCR (qRT-PCR) assays using β-actin as a control.

**Extraction of mRNA.** Prior to extraction, work surfaces, gloves, and instruments were treated with RNase-Zap and RNase-free water. Archived tissue was thawed on ice and then homogenized using TRIzol reagent followed by the addition of chloroform. Samples were mixed vigorously and centrifuged at 12,000g for 15 minutes at 4˚C. Supernatant was collected and washed with isopropyl alcohol and then centrifuged at 10,000g for 10 minutes at 4˚C. Supernatant was discarded and the remaining pellet was washed with 70% ethanol. Samples were centrifuged at 7500g for 5 minutes at 4˚C, the supernatant was discarded, and the pellet was re-suspended in RNase-free molecular grade water. Nucleic acid was quantified immediately using a NanoDrop. Samples were stored at -80˚C for later use.

**Synthesis of cDNA library.** Samples (DNase-treated RNA containing a cDNA master mix) were heated for 10 minutes at 25˚C followed by 120 minutes at 37˚C and finally 85˚C for one minute to terminate the reaction to complete the synthesis of cDNA.

**Gene expression assays.** Expression of WT-1, p53, CA-125, PAX2, and Ki67 genes in normal fimbriae, fimbriae with tumors, and ovaries with tumor were examined by reverse-transcriptase polymerase chain reaction (RT-PCR) and quantitative RT-PCR (qRT-PCR) assays using β-actin as a control. Synthesized cDNA was added to a RT-PCR master mix and placed in a thermocycler. Samples were heated for 10 minutes at 94˚C. This was followed by a cycling between denaturation (30 seconds at 95˚C), annealing (1 minute at 60˚C), and elongation (72˚C for 1 minute) for 25–35 cycles. The reaction was terminated after 10 minutes at 72˚C and samples were kept at 4˚C for later use. Agarose gel electrophoresis was performed at 100V for ~40 minutes. The gel was stained with an ethidium bromide solution, washed with deionized water, and imaged. Images were archived for later use. The Applied Biosystems ViiA7 Real-Time PCR system and associated computer software were utilized for qRT-PCR analysis (Thermo Fisher Scientific, Waltham, MA). Previously synthesized cDNA were plated in a 96-well reaction plate using SYBR ® Green supermix and the appropriate primers, listed below. Amplification during the RT-PCR stage cycled 40 times between 10 seconds at 95˚C followed by 30 seconds at 60˚C. The melt curve stage immediately followed where samples were heated at 95˚C for 15 seconds, 60˚C for 1 minute, and finally 95˚C for 15 seconds to terminate the reaction. Results were subjected to statistical analyses.

Following primers were used for hen tissues (*Gallus gallus*) (5' → 3'):

WT-1 **F:** CTGAAACGGCACCAAAGACG, **R:** ACCTGTATGAGTCGTGGTATGA;

p53 **F:** GCCGTGGCCGTCTATAAGAAA, **R:** CGGAAGTTCTCCTCCTCGATC;

CA-125 **F:** GGACCAGCCTCTACGTCAAC, **R:** TGCAGATGCTGTCTACTGCTG;

PAX2 **F:** GGCAGCAGGAAGCGACTATT, **R:** TCGCCCTTGATGTGGAACTG;

Ki67 **F:** CAGCAGATGTCTTAGCACCCA, **R:** TGCCTCCTCCATCAAGTTCTTC;

β-actin **F:** TGGCAATGAGAGGTTCAGGT, **R:** ATGCCAGGGTACATTGTGGT.

## Statistical analysis

Significant differences in the intensity of immunohistochemical expression of markers (WT-1, p53, CA-125, PAX2, Ki67) or gene expression among different groups were determined by ANOVA and independent sample t-test using GraphPad Prism version 6.0 (GraphPad Software Inc., La Jolla, CA). Differences were considered significant when *P<0.05.*

## Results

### Gross presentation

During gross examination following euthanasia, hens were grouped as: (1) healthy without any abnormality, (2) hens with solid masses in the fimbria and/or other parts of the infundibulum of the oviduct as well as in a part of the ovary, (3) solid mass only in the ovary without the involvement of the fimbria of the oviduct, or (4) solid masses in the ovary, fimbria and metastasized to other distant organs. A normal fully functional ovary in a regularly laying hen contains a hierarchy of 5 to 6 large pre-ovulatory follicles protruding from the ovarian surface (F1, F2 so on, Fig 1A). A normally laying hen lays an egg/day consecutively for 5 to 6 days and after taking one day rest, it resumes laying. In contrast, ovaries and oviducts in hens with ceased egg laying were regressed in size and showed no preovulatory follicles. Functionally, these hens resemble postmenopausal women (Fig 1B). Solid masses in all hens were accompanied with varying degrees of ascites, including little, moderate, or profuse amount, depending on the degree of the tumor metastasis. Based on the gross examination, hens with solid mass or masses in groups 2 and 3 were categorized as early stage OVCA and hens in group 4 were considered late stage OVCA (Fig 1C and 1D) (S2 Table). Of the total 135 hens examined, solid masses were detected in 27 hens. Of the 27 hens with tumors, solid masses in the fimbria and/or other parts of the infundibulum of the oviduct as well as in a part of the ovary in 7 hens (categorized as group 2) while solid masses in 8 hens were limited to the hen ovary (categorized as group 3) and in the remaining 12 hens, solid masses were detected in ovaries and/or fimbriae as well as in other distal organs (categorized as group 4).

The oviduct in a laying hen consists of five grossly and physiologically distinct parts including (1) infundibulum, (2) ampulla or magnum, (3) isthmus, (4) uterus or shell gland, and (5) vagina. The fimbria is the innermost part of the infundibulum located close to the ovary. Similar to women, fimbriae in hens appear like long, slender, finger-like projections known as the funnel of the infundibulum (Fig 2A). The fimbria and the remaining parts of infundibulum were studied separately. For fimbria, the cranial part of the infundibulum (approximately 2mm from the tip/edge of the fimbrial lip) were excised as reported by other investigators [36]. Solid masses were localized in the fimbria and adjacent areas in the infundibulum (Fig 2C).

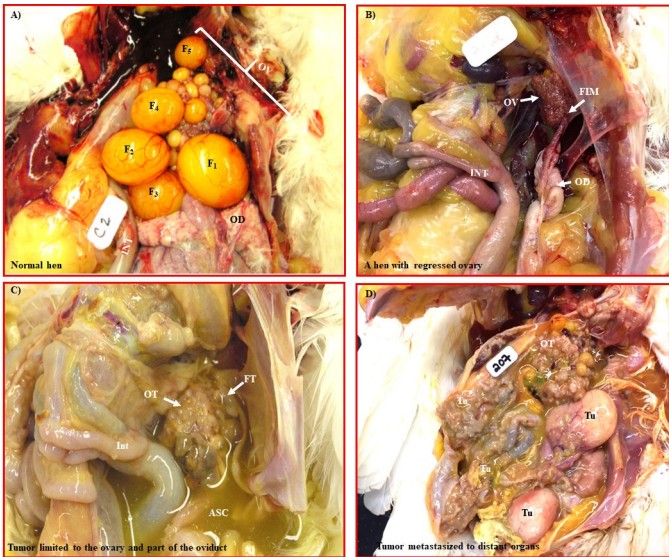

**Fig 1. Gross presentation of normal ovaries and ovaries with tumors in laying hens.** A) A normal fully functional ovary in a healthy laying hen showing a hierarchy of preovulatory follicles ($F_1$-$F_5$). B) A normal ovary in an older hen with ceased laying activity. The ovary is regressed, no preovulatory follicle is seen in the ovary and no abnormalities are observed at gross. Functionally, this ovary resembles a postmenopausal ovary in a woman. C) Tumor in ovary involving the fimbria of the infundibulum of the oviduct. The tumor is at early stage and is accompanied with moderate ascites. D) Ovarian cancer (OVCA) at late stage in a hen. Compared with early stage, the tumor in late stage showed extensive metastasis with solid masses formed in distant organs. ASC = Ascites, $F_1$ to $F_5$ = Hierarchical follicles: $F_1$ = The largest follicle destined to ovulate soon followed by $F_2$, $F_3$, so on, FIM = Fimbria, FT = Tumor in fimbria, INT = Intestine, OD = Oviduct, OV = Ovary, OT = Ovarian tumor, Tu = Tumor.

## Microscopic features

As observed in healthy subjects (S1 Fig), the fimbria in healthy hens consists of two surface epithelial layers (visceral and parietal), a muscular layer and the lumen (Fig 2). The surface epithelial layer consists of ciliated and taller columnar cells. Non-ciliated secretory goblet cells were seen to be intermingled with ciliated cells. The mucosal layer formed radial ridges and folds.

Histologically, the tumors were classified as serous, endometrioid, mucinous and mixed adenocarcinoma based on microscopic examination as reported earlier. Tumors in fimbriae showed papillary structures and marked nuclear atypia (Fig 2D and 2E) as seen in ovarian HGSC in humans (S1 Fig). The architecture was characterized by a lace-like papillary folding composed of cells with large pleomorphic nuclei containing mitotic bodies (red arrows indicate examples). These tumors were classified as serous-like carcinoma at early stage (7 hens in group 2). Of the 8 tumors limited to the ovary only (group 3), 6 showed confluent back-to-back glands surrounded by a single layer of malignant cells and were classified as endometrioid carcinoma. Tumor in one hen in this group showed malignant glands lined by columnar and goblet cells containing mucinous material in their lumen and was classified as mucinous carcinoma while the remaining one was a mixed sero-mucinous carcinoma. Of the 12 hens in group 4 with late-stage tumor, 5 were serous, 5 were endometrioid, one was mucinous, and one was poorly differentiated carcinoma, probably serous.

## Expression of WT-1

WT-1 expression in the surface epithelial cells of fimbria or ovary in healthy hens was either occasional or very weak (Fig 3A). No staining for WT-1 was observed in the stromal area of tumor tissues. In contrast, intense expression for WT-1 by the nuclei of malignant cells was

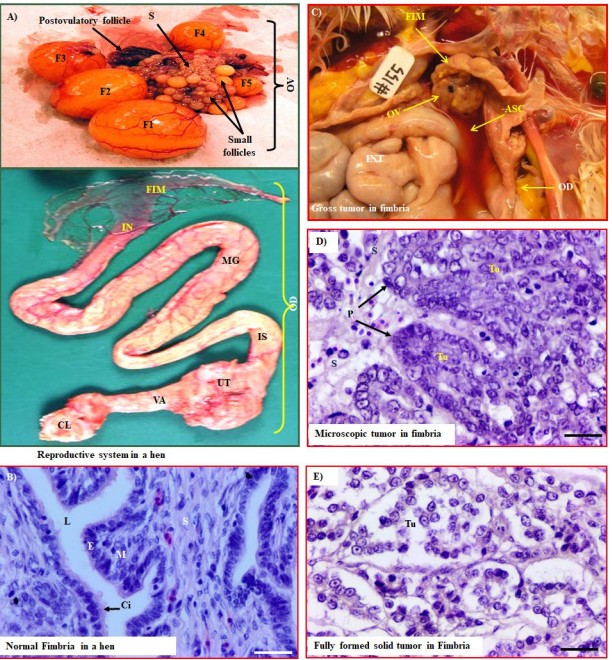

**Fig 2. Spontaneous malignant transformation of chicken oviductal fimbria. A)** Gross presentation of the reproductive system in a hen showing the ovary and different parts of the oviduct including oviductal fimbria (FIM), infundibulum (IN), magnum (MG), isthmus (IS), uterus (UT), vagina (VA). **B)** Microscopic presentation of chicken normal fimbria showing morphological similarities with that of human. **C)** Gross presentation of fimbrial tumor at early stage accompanied with ascites (ASC). **D)** Section of a microscopic tumor in fimbria showing papillary-like structure (P). **E)** Section of a fully formed tumor in fimbria. Ci = Cilia, CL = Cloaca, E = Surface epithelial layer, $F_1$ to $F_5$ = Hierarchical follicles, INT = Intestine, L = Lumen, M = Mucosa in fimbria, OV = Ovary, OD = Oviduct, S = Stroma, Tu = Tumor. Scale bar = 20μm.

detected in hens with fimbrial tumor and hens with tumors metastasized to distant organs (late stage OVCA) (Fig 3B and 3C). Similar patterns of WT-1 staining were detected in ovarian HGSC in patients (used as reference specimen) (S1 Fig).

Compared to the normal fimbria ($1.08 \times 10^4 \pm 1.6 \times 10^3$, mean ± SE in 20,000 μm$^2$ area of the tissue), the intensity of WT-1 expression was significantly higher in fimbriae with tumors ($1.58 \times 10^4 \pm 2.5 \times 10^3$, mean ± SE) in 20,000 μm$^2$ area of the tumor (*P < 0.05)* (Fig 3D). The intensity of WT-1 expression increased further to $3.04 \times 10^4 \pm 3.1 \times 10^3$ (mean ± SE) in 20,000 μm$^2$ area of the tumor in late stage OVCA (*P < 0.01)*.

Immunoblotting detected an immunoreactive band of approximately ~65kDa for WT-1. The pattern of intensity of WT-1 protein expression was similar as observed in immunohisto-chemical study including high expression in fimbrial tumor which increased further in hens with late stage OVCA (Fig 3E). Fimbrial and ovarian tissues from healthy hens showed faint expression for WT-1. Gene expression study also showed increase in WT-1 expression in association with malignant development in the oviductal fimbria in hens. Semi-quantitative RT-PCR assays showed a band for the WT-1 gene with high intensity in specimens from tumor-containing fimbriae as compared with the fimbriae of healthy hens (Fig 3F). Similarly, compared with normal fimbriae, significant (*P<0.01*) increase in WT-1 gene expression was observed in qRT-PCR in fimbriae with tumors and in late stage OVCA (Fig 3G). These results confirm the changes in the expression of WT-1 during tumor development in the fimbria as detected by immunoreactivity.

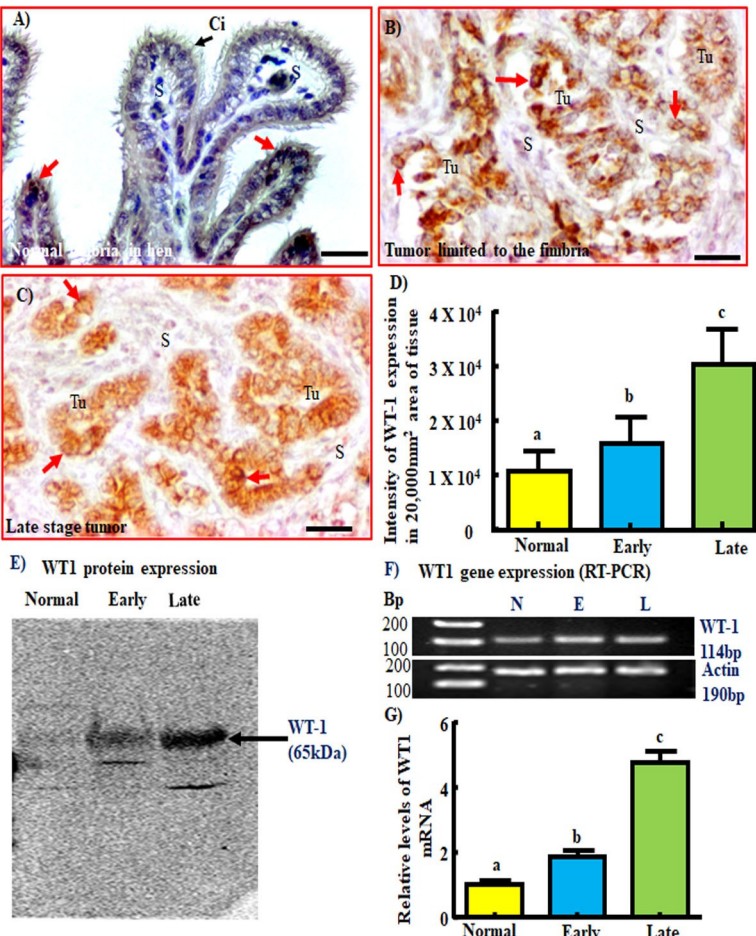

**Fig 3. Changes in WT-1 expression during malignant transformation and tumor development in oviductal fimbriae in hens.** A) Normal fimbria in a healthy hen showing very weak or occasional staining for WT-1 by the apical surface epithelial cells (red arrows are examples). B and C) Fimbria with tumor at early and late stages. In contrast to normal fimbria, malignant cells in fimbriae with tumors showed intense staining for WT-1. D) Intensity of immunohistochemical expression of WT-1. Compared with normal fimbria, the intensity of WT-1 was significantly higher in fimbria with tumor at early stage and increased further in late stages ($P < 0.001$). E) Immunoblotting showed a band of 65kDa for WT-1. The signal for WT-1 expression was very weak in normal fimbria while it was stronger in fimbria with tumor at early stage and strongest in late stage tumor. F and G) semi-quantitative (F) and quantitative (G) gene expression assays showed similar patterns of changes in WT-1 gene expression as observed in immunohistochemistry and immunoblotting. Bars with different letters are significantly different. Intensity or fold change values in (D) and (G), respectively, are arbitrary values presented as mean ± SEM. Bp = Base pair, Ci = Cilia, E = Early, L = Late, N = Normal, S = Stroma, Tu = Tumor. Scale bar = 20μm.

## Expression of p53

Expression of p53 in all specimens including normal fimbriae from healthy hens, WT-1-expressing fimbrial tumors, hens with tumors metastasized to other organs (late stage), as well as ovarian HGSC from patients (as reference specimens) was examined using an anti-p53 antibody (Millipore Sigma, St. Louis, MO) developed to detect expression of mutant p53 gene in humans. No staining for p53 was observed in normal fimbriae from healthy hens (Fig 4A). In contrast, abnormal staining for p53 was detected in the nuclei of malignant cells in all tumor sections, including those limited to the fimbria (early stage) and those at late stage showed abnormal staining for p53 (Fig 4B and 4C). The patterns of abnormal staining of p53

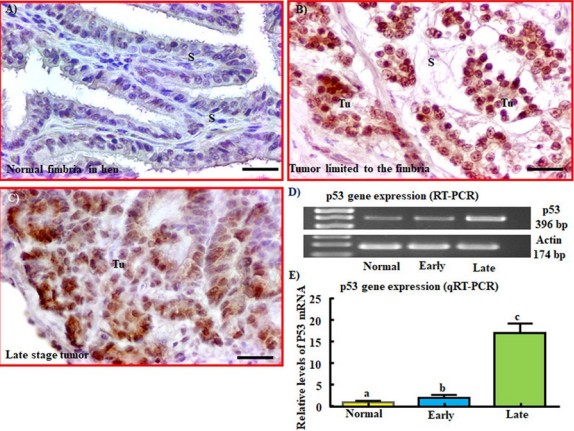

**Fig 4. Detection of abnormality in p53 protein and gene expression in fimbriae with tumor in hens.** A) Section of a normal fimbria in a hen. No staining for p53 is seen in normal fimbria. B and C) Section of a fimbria with tumor at early stage (B) and late stage (C) showing abnormal staining for p53. D) and E) Expression of p53 gene in normal fimbria and fimbria with tumor as detected by semi-quantitative (D) and quantitative (E) PCR (RT-PCR and qRT-PCR, respectively). Gene expression assays detected abnormal expression of p53 gene during malignant transformation which increased further as the tumor progressed to late stages ($P<0.001$). Fold change values are arbitrary values presented as mean ± SEM. Bars with different letters are significantly different. S = Stroma, Tu = Tumor. Scale bar = 20μm.

expression in these tumors was similar to the mutant p53 expression in ovarian HGSC in patients (used as reference specimens) (S1 Fig).

Quantitative and semi quantitative PCR assays were performed for expression of p53 gene using WT-1-expressing tumors limited to fimbriae (early stage) and hens with tumors metastasized to other organs (late stage) (Fig 4E and 4F). Quantitative PCR assays showed abnormal expression of p53 gene in tumor limited to the fimbria (early stage) which was stronger in late-

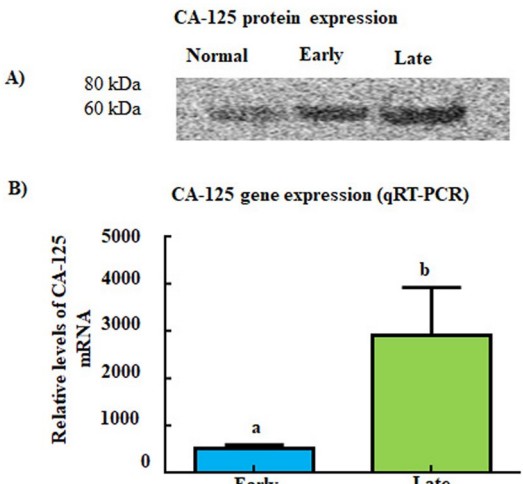

**Fig 5. Changes in CA-125 expression during the development and progression of tumor in oviductal fimbriae in hens.** A) Immunoblotting showed weak expression of CA-125 in normal fimbria. Compared with normal fimbria, strong signal for CA-125 was detected in fimbria with tumor at early stage and the signal was stronger in late-stage tumor. B) Quantitative gene expression (qRT-PCR) assays did not show significant increase in CA-125 levels with the progression of the disease to the late stages (P = 0.075). Fold change values are arbitrary values presented as mean ± SEM.

stage tumors (*P< 0.0001*). Similar patterns of abnormal expression of p53 gene (396 base pair) was observed in semi-quantitative PCR.

## Expression of CA-125

Strong staining for CA-125 was observed in tumors limited to the fimbria and tumors metastasized to distant organs (early and late stages, respectively). However, no staining for CA-125 was observed in the stroma of tumors. Strong immunoreactive band of approximately 70 kDa was seen for CA-125 in tumors limited to the fimbria (early stage) and tumor at late stages while it was weaker for fimbriae from healthy hens (Fig 5A). Quantitative gene expression assays (qRT-PCR) showed the difference in expression of CA-125 gene in fimbriae with tumors (early stage) and in ovarian tumors at late stage is not significant (*P = 0.075*) (Fig 5B). Similar patterns of expression for CA-125 were seen in ovarian HGSC from patients (used as reference).

## Expression of PAX2

Strong expression for PAX2 was observed in normal fimbriae from healthy hens (Fig 6). In contrast, the expression of PAX2 was remarkably reduced in the nucleus in malignant cells in tumors limited to the fimbria (early stage) and in tumors metastasized to distant organs (late stage) (Fig 6A–6C). The intensity of immunohistochemical expression of PAX2 in normal fimbriae was $1.64 \times 10^4 \pm 0.13 \times 10^4$ in 20,000 $\mu m^2$ area of tissue and decreased significantly in tumor limited to the fimbria (early stage) (*P<0.0001*) and in tumors metastasized to distant organs (late stage) (*P < 0.0001*) to $5.50 \times 10^3 \pm 0.87 \times 10^3$ and $1.26 \times 10^3 \pm 0.24 \times 10^3$, respectively, in 20,000 $\mu m^2$ area of tissue in (Fig 6D).

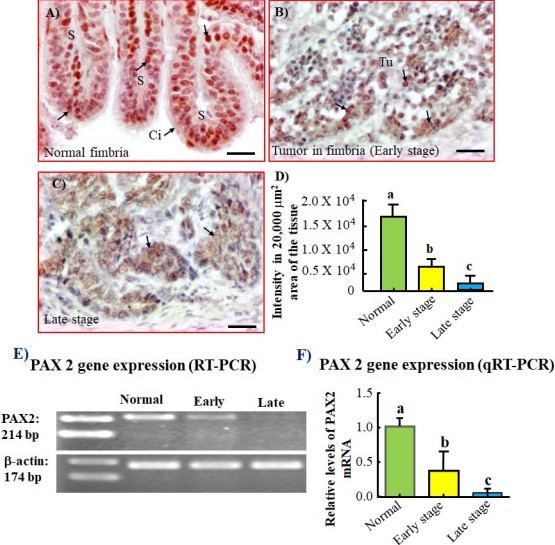

**Fig 6. Changes in PAX2 expression during tumor development in oviductal fimbriae in hens.** A) Section of a normal fimbria in a healthy hen showing strong staining for PAX2 by the surface epithelial cells. (B and C) Sections of fimbriae with tumor at early (B) and late (C) stages. In contrast to normal fimbria, malignant cells in fimbriae with tumors showed weaker staining for PAX2. (D) Intensity of immunohistochemical expression of PAX2. Compared with normal fimbria, the intensity of PAX2 was significantly lower in fimbriae with tumor at early stage and decreased further in late stage (*P<0.0001*). (E and F) semi-quantitative (E) and quantitative (F) gene expression assays showed significant decrease in PAX2 expression during tumor development and progression (*P<0.001*). Bars with different letters are significantly different. Intensity or fold change values in (D) and (F), respectively, are arbitrary values presented as mean ± SEM. Ci = Cilia. S = Stroma, Tu = Tumor. Scale bar = 20μm.

Quantitative and semi-quantitative PCR assays showed strong signal for expression of PAX2 in normal fimbriae from healthy hens. However, PAX2 expression declined gradually in association with malignant transformation (fimbriae with tumors) and was lost in late stage OVCA (Fig 6E and 6F).

## Expression of Ki67

Immunopositive Ki67 cells were localized both in normal fimbria and fimbria with tumors (S2 Fig). However, approximately 27% cells of normal fimbrial epithelium were found positive for Ki67 expression. In contrast, more than 90% malignant cells showed strong staining for Ki67 in tumors limited to the fimbria (early stage) and tumors metastasized to distant organs (late stage). Semi-quantitative PCR assays also showed similar patterns of gene expression for Ki67 in normal fimbriae and in specimens with tumors limited to the fimbria or tumors metastasized to distant organs (S2 Fig).

## Discussion

This is the first study reporting malignant transformation of the oviductal fimbria in laying hens, a preclinical model of spontaneous OVCA. Fallopian tube (Oviduct) fimbria in women has recently been suggested to be the most common site of origin of ovarian high grade serous carcinoma (HGSC), the highly lethal form of OVCA. Difficulty in accessing the fimbrial tissues in women to study the pathophysiology of ovarian HGSC necessitates the exploration of a preclinical model with oviductal fimbria that develops spontaneous OVCA. The current study was undertaken to examine the incidence of malignant transformation in the oviductal fimbria in laying hens. This study revealed that oviductal fimbriae in hens develop malignant tumors which histologically resemble ovarian HGSC-like features in patients. These tumors expressed several markers including WT-1, p53, CA-125, PAX2 and Ki67 similarly observed in ovarian HGSC. Thus, the fimbria of the laying hen offers a potential model to study spontaneous malignant transformation leading to ovarian HGSC and to develop an early detection test and interventions against it.

In this study, routine staining with hematoxylin and eosin showed malignant cells with large pleomorphic nuclei containing mitotic bodies arranged in a papillary-like structure in some tumors limited to fimbriae (early stage). Other tumors showed masses like a compact sheet of malignant large and pleomorphic cells. Back-to-back tumor glands separated by fibro-muscular tissue were seen to be arranged in the tumor stroma. Similar microscopic features have been reported in hens with OVCA and ovarian HGSC in patients earlier [15].

The Wilms Tumor 1 (WT-1) gene encodes a transcription factor that is involved in the regulation of cellular development and survival including in the urogenital system [37]. WT-1 is necessary for the development of critical organs, including the kidneys, and also plays a role as a tumor suppressor against the formation of Wilms' tumor, a cancer of the kidneys [38]. Enhanced expression of WT-1 has been reported in colorectal, breast, and epithelial ovarian cancers [39]. In this study, intense expression of WT-1 by the nucleus of malignant cells was detected in tumors limited to the fimbriae. However, no staining was observed in normal fimbriae from healthy hens, suggesting an association of WT-1 expression with malignant transformation. Similar patterns of intense expression of WT-1 by the nucleus of the malignant cells were also observed in ovarian HGSC in women. The expression of WT-1 increased further in specimens with tumors at late stage. Enhancement in WT-1 expression was also observed in immunoblotting and genomic studies. In this study, a band of approximately 65 kDa was detected for WT-1 in tumor specimens. WT-1 has been reported to be expressed at multiple band sizes, ranging from 36–65 kDa in chicken [40] and human [41, 42]. Thus, these

results suggest that fimbrial tissues in hens may be a site of ovarian HGSC-like malignant development.

Wildtype TP53 is a well-known tumor suppressor that effectively prevents the development of many cancers. A major role of p53 in preventing tumor development is to prevent DNA damage and cellular stress [43]. In contrast, mutant TP53 favors oncogenic pathways and promotes tumor growth by dysregulation of DNA repair mechanisms and resisting apoptosis or cellular senescence [44]. TCGA data reported mutation in TP53 gene in over 96% of ovarian HGSC cases [45]. This study showed abnormal expression of p53 by the malignant cells in tumors limited to the fimbria in hens (early stage) while no expression for p53 was recorded in normal fimbrial tissues from healthy hens. These observations were supported by the immunoblotting and gene expression assays for p53. Similarities in the patterns of abnormal expression of p53 in tumors limited to the fimbriae in hens and expression of mutant p53 in ovarian HGSC in patients suggest the suitability of fimbrial tissues in chickens as a model to study ovarian HGSC.

Serum CA-125 level is a currently used marker for the detection of OVCA [46]. CA-125 levels are also commonly elevated in endometriosis, infection, and benign tumors. As a biomarker CA-125 has high specificity but low sensitivity for OVCA, especially in detecting early stage OVCA [47]. The exact biological role of CA-125 has yet to be elucidated, but it has been proposed that this antigen interacts with different aspects of the immune system, like NK cells, and may potentially be involved in the immune evasion of tumors in OVCA progression [48]. The association of CA-125 and OVCA is well-reported and the specificity of CA-125 makes it a suitable choice for demonstrating the histological similarities in HGSC of the oviductal fimbria in hens and ovarian HGSC in women. This study showed strong expression of CA-125 in hens with tumor limited to the fimbria (early stage) and tumor at late stage as observed in ovarian HGSC in patients. Thus, enhanced expression of CA-125 during malignant transformation and similarities in the expression patterns to ovarian HGSC in patients further supports the suitability of oviductal fimbria in hens as a model to study ovarian HGSC.

This study showed compared with normal fimbria, expression of PAX2 protein as well as PAX2 gene was lower in fimbria with tumor limited to the fimbria and the expression was lowest in late stage tumors [49]. PAX2 has been shown to be expressed in normal adult female reproductive tissues [50], be absent in ~85% of HGSC, and be expressed in clear cell and mucinous tumors [30, 49]. Thus, this result supports the previous report [18] that a portion of epithelial ovarian tumors in hens also express PAX2, an oviductal protein, and its expression decreases with the progress of the OVCA.

Ki67 is a marker of proliferation and growth of cells and can be specifically used as a marker of tumor growth since it is only detected in dividing cells [51]. While it is not specific to ovarian HGSC, it is indicative of malignancy because of the rapid growth rate observed during malignant transformation. Thus, high prevalence of intense Ki67 expression supports the incidence of malignant change. It has been suggested by other studies that Ki67 may also serve as a marker of metastasis in cancer, further promoting its role as a prognostic marker in a variety of cancers [52]. The intense immunohistochemical staining for Ki67 observed in fimbrial tumors in hens as well as the results of genomic assays support the notion of malignant-associated cellular growth in the fimbria, as seen in ovarian HGSC in patients.

Taken together, the results of the present study suggest that the oviductal fimbria in hens is a site of malignant transformation. The histological features of fimbrial tumors in hens closely resemble the microscopic features of ovarian HGSC in patients and show similarities in expression of several putative molecular markers. Therefore, the oviductal fimbria in laying hens offers a potential preclinical model to establish an early detection test for ovarian HGSC and to develop targeted therapeutics, including immunotherapeutics, as well as prevention

strategies. Fimbrial tumors in hens may also be a preclinical model to develop a targeted ultra-sound imaging method, as reported earlier [53], to detect malignant-associated changes relative to ovarian HGSC.

Small sample size may be a limitation of this study. Moreover, a lack of information on serum levels of the markers may also be a significant limitation of the study. However, the results presented here including the histological features and expression of markers as detected in immunohistochemical, immunoblotting and gene expression assays will serve as a foundation for serum assays and early detection of the disease.

## Conclusion

In conclusion, this study showed that oviductal fimbrial epithelium in laying hens may become malignant and develop tumor masses. These tumors showed similarities in their microscopic features and in the expression of molecular markers including WT-1, p53, CA-125, PAX2, and Ki67 to those observed in ovarian HGSC in women. Thus, the fimbrial tumors in hens may be a potential model to study to generate information on different aspects of ovarian HGSC.

## Supporting information

**S1 Fig. Sections of ovarian high grade serous carcinoma (HGSC) stained for expression of several markers used as positive control to study similarities in staining patterns between patients and hens.** (A) Section of a normal fimbria from a subject. Histological features are similar with the fimbria in hens shown in Fig 2. (B) Section of an ovarian HGSC showing papillary-like feature of the tumor surrounded by stroma. Such features were also seen in fimbria with tumor in hens (Fig 2). (C-F) Sections of ovarian HGSC stained for expression of WT-1, mutated p53, PAX2 and Ki67 staining, respectively. Similar patterns of staining for these markers were also observed in fimbriae with tumor in hens (presented in Figs 3–5).
E = Ovarian surface epithelial cells, S = Stroma, Tu = Tumor. Scale bar = 20μm.
(TIF)

**S2 Fig. Expression of Ki67 in normal oviductal fimbria and fimbriae with tumors at early and late stages in hens.** (A) Surface epithelial cells in normal fimbria (FSE) showed variable staining including strong staining by apical cells while basal cells showed weak staining for Ki67 expression. (B and C) In contrast to normal fimbria, intense staining for Ki67 staining was observed in fimbriae with tumor at early stage (B) and late stage (C). D) RT-PCR showing gene expression of Ki67 and keratin 8 was used as control. As in immunohistochemistry, strong signal for Ki67 in fimbriae with tumor was also observed in gene expression assay.
Ci = Cilia, M = Mucosa of the fimbria, S = Stroma, Tu = Tumor. Scale bar = 20μm.
(TIF)

**S1 Table. Information on primary antibodies used in this study.**
(DOCX)

**S2 Table. Distribution of hens with or without ovarian tumors based on their gross presentation.**
(DOCX)

**S1 Raw images. Original blot/gel image data used in this study.**
(PDF)

## Acknowledgments

The staffs of the Poultry Research Farm of the University of Illinois at Urbana-Champaign (UIUC), Urbana, Illinois and the assistance of Bill W. Hanafin, Laboratory Assistant, Bahr's Lab, Department of Animal Sciences, UIUC, Urbana, Illinois are acknowledged.

## Author Contributions

**Conceptualization:** Elizabeth A. Paris, Pincas Bitterman, Animesh Barua.

**Data curation:** Elizabeth A. Paris, Animesh Barua.

**Formal analysis:** Elizabeth A. Paris, Sanjib Basu.

**Funding acquisition:** Animesh Barua.

**Investigation:** Elizabeth A. Paris, Animesh Barua.

**Methodology:** Elizabeth A. Paris, Janice M. Bahr, Pincas Bitterman, Jacques S. Abramowicz, Animesh Barua.

**Project administration:** Animesh Barua.

**Resources:** Animesh Barua.

**Supervision:** Janice M. Bahr, Animesh Barua.

**Writing – original draft:** Elizabeth A. Paris.

**Writing – review & editing:** Janice M. Bahr, Pincas Bitterman, Sanjib Basu, Jacques S. Abramowicz, Animesh Barua.

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
