## [Decision Letter · Decision Letter 0]

11 May 2021

PONE-D-21-12611

Incidence of malignant transformation in the oviductal fimbria in laying hens, a preclinical model of spontaneous ovarian cancer

PLOS ONE

Dear Dr. Barua,

Thank you for submitting your manuscript to PLOS ONE. After careful consideration, we feel that it has merit but does not fully meet PLOS ONE’s publication criteria as it currently stands. Therefore, we invite you to submit a revised version of the manuscript that addresses the points raised during the review process.

We look forward to receiving your revised manuscript.

Kind regards,

Yihong Wang

Academic Editor

PLOS ONE

Additional Editor Comments:

Based on the advice received, we felt that your manuscript could be accepted for publication should you be prepared to incorporate the reviewer’s questions/comments in the revision.

As the Academic Editor, I have a couple of general suggestions. First, please respect the reviewers by taking their suggestions seriously because they took the time and effort to provide their reviews. If you decide to revise and resubmit the manuscript, please include a sheet indicating: (1) how you followed the suggestions and (2) why you did not follow the suggestions. Please include specific citations to justify your responses if you disagree with suggestions from the reviewers.

Journal Requirements:

2. Please provide the product number and any lot numbers of the antibodies purchased for your study.

3.  At this time, we request that you  please report additional details in your Methods section regarding animal care, as per our editorial guidelines:

(1) Please state the source of hens used in the study

(2) Please provide details of animal welfare (e.g., shelter, food, water, environmental enrichment), and any steps taken to minimize animal suffering and distress

(3) Please include the method of euthanasia

Thank you for your attention to these requests.

4. In your ethics statement in the manuscript and in the online submission form, please ensure that you have discussed whether all data/samples were fully anonymized before you accessed them and/or whether the IRB or ethics committee waived the requirement for informed consent. If patients provided informed written consent to have data/samples from their medical records used in research, please include this information.

5. Thank you for including your ethics statement:  "Animals were used in this study as per the approved protocol of the IACUC, University of Illinois at Urbana-Champaign (UIUC). The approved number is 19091. ".   

Please amend your current ethics statement to confirm that your named institutional review board or ethics committee specifically approved this study

Reviewers' comments:

Reviewer's Responses to Questions

**Comments to the Author**

1. Is the manuscript technically sound, and do the data support the conclusions?

Reviewer #1: Partly

Reviewer #2: Yes

2. Has the statistical analysis been performed appropriately and rigorously? 

Reviewer #1: I Don't Know

Reviewer #2: Yes

3. Have the authors made all data underlying the findings in their manuscript fully available?

Reviewer #1: Yes

Reviewer #2: Yes

4. Is the manuscript presented in an intelligible fashion and written in standard English?

Reviewer #1: Yes

Reviewer #2: Yes

5. Review Comments to the Author

Reviewer #1: This manuscript communicates the very interesting occurrence of tumors located on the fimbriae of the laying hen. This is significant in that high grade serous cancer in women is thought to arise from a primary lesion in the fimbria, characterized by a p53 signature in the secretory cells. Other groups have identified commonalities in gene expression between hen ovarian tumors and normal oviduct (see Trevino, LS, Giles, JR Wang, W, Urick ME Johnson PA. Gene Expression Profiling Reveals Differentially Expressed Genes in Ovarian Cancer of the Hen: Support for Oviductal Origin? Hormones and Cancer. August 2010; 1(4):177-86 DOI: 10.1007/s12672-010-0024-8) and mutations in p53 have been identified by sequencing cDNA from ovarian tumors (reference 16, Hakim, et al). These observations are relevant to this manuscript but have been glossed over in the introduction (lines 94-96).

Until the authors are able to demonstrate that the p53 gene is mutated, they should refrain from referring to it as mutant and instead describe it as abnormal staining. The assumption that human p53 staining in tumors is the result of a mutation has been backed up innumerable times by sequencing the gene or transcript. Furthermore, the PAX family (2, 5 and 8) has been shown to inhibit p53 transcription, so the increase in p53 transcript may simply reflect loss of PAX2 protein expression (ref. Stuart ET, Haffner R, Oren M, Gruss P. Loss of p53 function through PAX-mediated transcriptional repression. EMBO J. 1995 Nov 15; 14(22):5638-45. PMID: 8521821; PMCID: PMC394679.)

Additionally, the Pax 8 gene is not present in avian species including chickens (ref Paixao-Cortes VR, Salzano FM, Bortolini MC (2013) Evolutionary History of Chordate PAX Genes: Dynamics of Change in a Complex Gene Family. PLoS ONE 8(9): e73560. doi:10.1371/journal.pone.0073560). Therefore it is recommended that the authors remove of all mention of PAX8.

The data on PAX8 protein raises the question of how all of the antibodies used were validated for use in the chicken. For example, the western blot of WT-1 in figure 3E shows the molecular weight to be 65 kDa whereas other references show chicken embryo WT-1 to have a molecular weight of 42 kDa (Cell Tissue Res (2001) 303:173–186 DOI 10.1007/s004410000307 R. Carmona · M. González-Iriarte J.M. Pérez-Pomares · R. Muñoz-Chápuli Localization of the Wilms’ tumour protein WT1 in avian embryos).

Additionally, the antibodies that were used in this manuscript should be identified by catalog number or clone.

The intensity of Ki67 expression in figure S2 seems relatively comparable between normal fimbria and the tumor cells, although normal fimbriae are described to have “few” Ki67 positive cells (page 18, line382). In fact, it is difficult to determine if any of the normal fimbrial epithelial cells are negative for Ki67. It would be more accurate to report the number of positive cells relative to the total number of epithelial cells in several fields. It is misleading to use actin as a normalization protein in the western blot as there are varying amounts of non-Ki67 positive tissues (actin-containing stroma) in the samples. An epithelial specific marker would more accurately reflect the Ki67 quantity in the epithelial compartments.

A table should be added that shows the number of hens in each category as described on page 10 in Gross Presentation. Hens categorized in group 4 had masses in ovaries and/or fimbriae (line 226 page 11) these should be separated into ovary, fimbria and both (three subcategories of group 4). It is not clear if all hens in group 2 had ovarian tumors, or also oviductal tumors. The use of and/or (page 10 line 211 and page 11 line 226) renders these categorizations very murky

If the study was undertaken to examine the incidence, (page 5, line 119 and page 18 line 394), what is the incidence of malignant transformation in the fimbriae? The answer might be more obvious by the inclusion of the table, if not explicitly stated.

Were patient specimens from BRCA1 mutated fimbrial tissue used in this manuscript? If not, should they be included in the description of clinical specimens (page 6 line 138)?

Figure 1 C is described as a tumor in the ovary involving the fimbria. An arrow pointing to this involvement would be useful. A suggestion for future photographs would be to remove the tumors and reproductive tract from the animal for clearer viewing.

Page 5 line 111. Should read Tumor protein p53, not tissue.

Reviewer #2: The aim of this study was to examine if the oviductal fimbria in hens is a site of origin of Ovarian high grade serous carcinoma (HGSC) and whether it expresses several putative markers expressed in ovarian HGSC in patients. Authors showed that, compared with normal fimbriae, intensities of expression of WT-1, mutant p53, CA-125, PAX8, PAX2 and Ki67 were higher in fimbrial tumors, however, the expression of PAX2 decreased gradually as the tumor progressed to late stages. The patterns of expression of these markers were similar to those in ovarian HGSC patients. Thus, authors suggest that tumors of the oviductal fimbria in hens may offer a preclinical model to study different aspects of spontaneous ovarian HGSC in women.

This study is designed with unique idea to use a chicken model that enables to investigate the factors responsible for the ovarian and infundibular cancer. The results show interesting and novel findings that expression profiles of specific markers were similar to those in ovarian HGSC patients, and thus tumors of the oviductal fimbria in hens may offer a preclinical model to study the spontaneous ovarian HGSC.

Major comments

1. Authors describe the examined tissue only as the infundibulum. However, the hen infundibulum has a unique structure; namely, at the edge of the fimbria (approximately 2 mm), mucosal surface on the inner side extends even on the outer side before the epithelium transit to serosa. Thus, both inner and outer surface of this part of the fimbria are surrounded serosa by mucosal epithelium. Furthermore, lymohocyst-like structures may appear occasionally in that outer mucosal epithelium. Authors show the micrographs for the immunohistochemistry; they should make it clear which parts of the infundibulum were examined, cephalic, middle, caudal, inner side or outer side.

The following paper may help the authors to understand the comments of this reviewer.

Fujii S et al. (!981) The regional morphology of the infundibulum of the hen's oviduct with special reference to the mechanism of the engulfing of the ovulated ovum. J. Fac. Appl. Biol. Sci. Hiroshima Univ. 20: 87-98. ( https://ir.lib.hiroshima-u.ac.jp/files/public/2/23426/20141016143413572892/24-2126.pdf )

2. In the Results section, Figure legends are presented. It makes difficult to read the manuscript, and thus the figure legends should be put on separate pages.

3. Quality of Fig. 2A is not acceptable. Authors should prepare a new photograph to replace with the current one.

Specific comments

L134: Give the name of the institute that approved the animal experiment.

L154: What is D2O? Add full spelling at the place where the abbreviation appears first.

L155: What is neutralization? You may mean the endogenous peroxidase was quenched.

L159-160: Add the name of animals used to raise the antibodies.

L172: Add how to confirm the specificity of the immunostaining. Although western blotting was performed, IHC specificity is also required.

L174-183: Is the word of “intensity of the staining” correct? This reviewer assumes it may the positive area in 20,000 μm2.

L193: Procedure for the total RNA extraction, revers transcription for CDNA synthesis, and qRT-PCR condition should be added.

L202: Actin should be β-actin.

L241: Delete cloaca, cloaca is not a part of oviduct.

L260: Crypts in the funnel part may not be the secretory glands, although the chalaza is secreted in the tubular region of the infundibulum.

L265: Specify the pleomorphic nuclei containing mitotic bodies by arrows on the micrographs.

L284: Specify the intense staining for WT-1 by arrows on the micrographs.

L344: Although authors describe the expression is significant, P=0.075 does not show significance.

L355: Specify the intense staining for PAX8 by arrows on the micrographs.

L459-450: This sentence is difficult to follow.

Figures: Magnification should be shown by scale bars. “X40” is not a correct magnification.

6. PLOS authors have the option to publish the peer review history of their article (what does this mean?). If published, this will include your full peer review and any attached files.

Reviewer #1: No

Reviewer #2: No

---

## [Author Response · Author response to Decision Letter 0]

25 Jun 2021

Responses to the Academic Editor:

We are thankful for your review of our manuscript and the specific recommendations provided to align with PLOS ONE’s requirements. The manuscript has been revised (specific changes listed below). 

1. PLOS ONE’s style requirements were reviewed and met. 

2. Information on antibodies in the study is included in S1 Table.

3. The source of hens used in the study, details of animal welfare, and methods of euthanasia are included in lines 132-138 in the revised version. 

4. Samples were fully anonymized prior to our access and prior to the start of this study. This information was included in lines 153-154 in the revised version. This statement will be uploaded to the online submission form.

5. Methods section was revised to state that protocol number 19091 was specifically approved by the Institutional Animal Care and Use Committee (IACUC) of the University of Illinois at Urbana-Champaign (UIUC) in lines 144-146. This statement will be uploaded to the Ethics Statement during submission. 

6. Original blot/gel image data are provided as supplementary information. 

Responses to the comments of Reviewer 1:

We are very thankful for your time to review our manuscript and appreciate your comments and suggestions on our manuscript. We have revised the manuscript in line with your comments and suggestions. Specific responses are mentioned below: 

Query #1 (Q#1): This manuscript communicates the very interesting occurrence of tumors located on the fimbriae of the laying hen. This is significant in that high grade serous cancer in women is thought to arise from a primary lesion in the fimbria, characterized by a p53 signature in the secretory cells. Other groups have identified commonalities in gene expression between hen ovarian tumors and normal oviduct (see Trevino, LS, Giles, JR Wang, W, Urick ME Johnson PA. Gene Expression Profiling Reveals Differentially Expressed Genes in Ovarian Cancer of the Hen: Support for Oviductal Origin? Hormones and Cancer. August 2010; 1(4):177-86 DOI: 10.1007/s12672-010-0024-8) and mutations in p53 have been identified by sequencing cDNA from ovarian tumors (reference 16, Hakim, et al). These observations are relevant to this manuscript but have been glossed over in the introduction (lines 94-96).

Response #1 (R#1): We agree with the reviewer that Trevino et al., (PMID: 21761365) has reported very important information that gene expression between ovarian tumors and normal oviduct in laying hens are similar. We have included this information in lines 95-98 of the revised version. 

Q#2A: Until the authors are able to demonstrate that the p53 gene is mutated, they should refrain from referring to it as mutant and instead describe it as abnormal staining. The assumption that human p53 staining in tumors is the result of a mutation has been backed up innumerable times by sequencing the gene or transcript. Furthermore, the PAX family (2, 5 and 8) has been shown to inhibit p53 transcription, so the increase in p53 transcript may simply reflect loss of PAX2 protein expression (ref. Stuart ET, Haffner R, Oren M, Gruss P. Loss of p53 function through PAX-mediated transcriptional repression. EMBO J. 1995 Nov 15; 14(22):5638-45. PMID: 8521821; PMCID: PMC394679.)

R#2A: As suggested, we have changed the statement “mutation in p53” to “abnormal staining of p53” throughout the text. Furthermore, additional relevant information on the anti-p53 antibodies is also included in lines 363-366 of the revised version. 

Q#2B: Additionally, the Pax 8 gene is not present in avian species including chickens (ref Paixao-Cortes VR, Salzano FM, Bortolini MC (2013) Evolutionary History of Chordate PAX Genes: Dynamics of Change in a Complex Gene Family. PLoS ONE 8(9): e73560. doi:10.1371/journal.pone.0073560). Therefore it is recommended that the authors remove of all mention of PAX8.

R#2B: As suggested, all mention of PAX8 has been removed. However, we felt it would be highly relevant to include additional information on PAX2. Thus, in the revised version, data from additional experimentation on PAX2 is added. Results = lines 404-427, Discussion = 495-501

Q#3: The data on PAX8 protein raises the question of how all of the antibodies used were validated for use in the chicken. For example, the western blot of WT-1 in figure 3E shows the molecular weight to be 65 kDa whereas other references show chicken embryo WT-1 to have a molecular weight of 42 kDa (Cell Tissue Res (2001) 303:173–186 DOI 10.1007/s004410000307 R. Carmona · M. González-Iriarte J.M. Pérez-Pomares · R. Muñoz-Chápuli Localization of the Wilms’ tumour protein WT1 in avian embryos).

R#3: WT-1 has been shown to exist in multiple isoforms ranging from 36-65 kDa and this has been stated in lines 467-469 of the revised version. 

In addition, validation of antibodies used is stated in lines 184-186 of the revised version. 

Furthermore, the antibodies that were used in this manuscript are identified by catalog number or clone in S1 Table. 

Q#4: The intensity of Ki67 expression in figure S2 seems relatively comparable between normal fimbria and the tumor cells, although normal fimbriae are described to have “few” Ki67 positive cells (page 18, line382). In fact, it is difficult to determine if any of the normal fimbrial epithelial cells are negative for Ki67. It would be more accurate to report the number of positive cells relative to the total number of epithelial cells in several fields. It is misleading to use actin as a normalization protein in the western blot as there are varying amounts of non-Ki67 positive tissues (actin-containing stroma) in the samples. An epithelial specific marker would more accurately reflect the Ki67 quantity in the epithelial compartments.

R#4: As suggested, the number of Ki67-positive cells was counted and reported in lines 430-432 of the revised version. In addition, the micrograph of normal fimbria in S2 Fig A has been replaced to better represent the observed Ki67 expression.

The RT-PCR (mistaken as a western blot in this comment) was re-done using keratin 8 as the control for each sample. Keratin 8 is an epithelial marker and thus serves as a positive control for Ki67 gene expression. This is found in S2 Fig D.

Q#5: A table should be added that shows the number of hens in each category as described on page 10 in Gross Presentation. Hens categorized in group 4 had masses in ovaries and/or fimbriae (line 226 page 11) these should be separated into ovary, fimbria and both (three subcategories of group 4). It is not clear if all hens in group 2 had ovarian tumors, or also oviductal tumors. The use of and/or (page 10 line 211 and page 11 line 226) renders these categorizations very murky. If the study was undertaken to examine the incidence, (page 5, line 119 and page 18 line 394), what is the incidence of malignant transformation in the fimbriae? The answer might be more obvious by the inclusion of the table, if not explicitly stated.

R#5: As suggested, a table showing the number of hens in each category as described in gross presentation is resented in S2 Table and mentioned in line 269. 

Q#6: Were patient specimens from BRCA1 mutated fimbrial tissue used in this manuscript? If not, should they be included in the description of clinical specimens (page 6 line 138)?

R#6: Removed from the revised version. 

Q#7: Figure 1 C is described as a tumor in the ovary involving the fimbria. An arrow pointing to this involvement would be useful. A suggestion for future photographs would be to remove the tumors and reproductive tract from the animal for clearer viewing.

R#7: An arrow has been added to Fig 1C in the revised version. 

Q#8: Page 5 line 111. Should read Tumor protein p53, not tissue.

R#8: Corrected in line 113 of the revised version.

Responses to the comments of Reviewer 2:

We are very grateful for your time to review our manuscript and appreciate your suggestions. We have revised the manuscript according to your comments and suggestions. Specific responses are mentioned below:

Major comments

Q#1: Authors describe the examined tissue only as the infundibulum. However, the hen infundibulum has a unique structure; namely, at the edge of the fimbria (approximately 2 mm), mucosal surface on the inner side extends even on the outer side before the epithelium transit to serosa. Thus, both inner and outer surface of this part of the fimbria are surrounded serosa by mucosal epithelium. Furthermore, lymohocyst-like structures may appear occasionally in that outer mucosal epithelium. Authors show the micrographs for the immunohistochemistry; they should make it clear which parts of the infundibulum were examined, cephalic, middle, caudal, inner side or outer side.

The following paper may help the authors to understand the comments of this reviewer.

Fujii S et al. (!981) The regional morphology of the infundibulum of the hen's oviduct with special reference to the mechanism of the engulfing of the ovulated ovum. J. Fac. Appl. Biol. Sci. Hiroshima Univ. 20: 87-98. (https://ir.lib.hiroshima-u.ac.jp/files/public/2/23426/20141016143413572892/24-2126.pdf )

R#1: More information on the fimbria of the infundibulum used in this study (cranial part) is included in lines 290-292 of the revised version. 

Q#2. In the Results section, Figure legends are presented. It makes difficult to read the manuscript, and thus the figure legends should be put on separate pages.

R#2: This is a format-related requirement of the journal – no change is added in the revised version. 

Q#3. Quality of Fig. 2A is not acceptable. Authors should prepare a new photograph to replace with the current one.

R#3: As suggested, part of Fig 2A has been modified in the revised version. 

Specific comments:

L134: Give the name of the institute that approved the animal experiment. 

Listed in line 145-146 of the revised version.

L154: What is D2O? Add full spelling at the place where the abbreviation appears first.

Defined in line 165 of the revised version.

L155: What is neutralization? You may mean the endogenous peroxidase was quenched.

Wording was modified to the suggested “quenching” in line 167 of the revised version

L159-160: Add the name of animals used to raise the antibodies.

Animals used to raise antibodies included in S1 Table 

L172: Add how to confirm the specificity of the immunostaining. Although western blotting was performed, IHC specificity is also required.

Confirmation of immunostaining specificity described in lines 184-186 of the revised version.

L174-183: Is the word of “intensity of the staining” correct? This reviewer assumes it may the positive area in 20,000 μm2.

The phrase “intensity of the staining” is correct for the purposes of our analyses. No changes have been made. 

L193: Procedure for the total RNA extraction, revers transcription for CDNA synthesis, and qRT-PCR condition should be added.

Details of RNA extraction, reverse-transcription for cDNA synthesis, and conditions for (q)RT-PCR have been described in lines 210-241 of the revised version.

L202: Actin should be β-actin.

Correction to β-actin completed in line 248 of the revised version.

L241: Delete cloaca, cloaca is not a part of oviduct.

Cloaca has been removed. 

L260: Crypts in the funnel part may not be the secretory glands, although the chalaza is secreted in the tubular region of the infundibulum.

Part of that line has been removed of the revised version (lines 308-309). 

L265: Specify the pleomorphic nuclei containing mitotic bodies by arrows on the micrographs.

Arrows have been added to S1Fig and mentioned in line 314 of the revised version.

L284: Specify the intense staining for WT-1 by arrows on the micrographs.

Arrows have been added to Fig 3.

L344: Although authors describe the expression is significant, P=0.075 does not show significance.

Expression was mistakenly listed as significant. Lines 395 and 401 were corrected accordingly in the revised version. 

L355: Specify the intense staining for PAX8 by arrows on the micrographs.

PAX8 has been removed from the manuscript. Arrows have been added to Fig6 of the revised version to indicate intense PAX2 staining. 

L459-460: This sentence is difficult to follow.

This sentence has been re-written in lines 505-506 of the revised version. 

Figures: Magnification should be shown by scale bars. “X40” is not a correct magnification.

Figures contain scale bars when appropriate and figure legends include size of scale bar.

---

## [Editor Report · Decision Letter 1]

8 Jul 2021

Incidence of malignant transformation in the oviductal fimbria in laying hens, a preclinical model of spontaneous ovarian cancer

PONE-D-21-12611R1

Dear Dr. Barua,

We’re pleased to inform you that your manuscript has been judged scientifically suitable for publication and will be formally accepted for publication once it meets all outstanding technical requirements.

Kind regards,

Yihong Wang

Academic Editor

PLOS ONE

Additional Editor Comments (optional):

All the questions and comments have been addressed.
---

## [Editor Report · Acceptance letter]

12 Jul 2021

PONE-D-21-12611R1 

Incidence of malignant transformation in the oviductal fimbria in laying hens, a preclinical model of spontaneous ovarian cancer 

Dear Dr. Barua:

I'm pleased to inform you that your manuscript has been deemed suitable for publication in PLOS ONE. Congratulations! Your manuscript is now with our production department. 

Kind regards, 

on behalf of

Dr. Yihong Wang 

Academic Editor

PLOS ONE